# Challenges in the Detection of Polymyxin Resistance: From Today to the Future

**DOI:** 10.3390/microorganisms12010101

**Published:** 2024-01-04

**Authors:** Rebeca Siqueira Rubens, Isabel de Souza Andrade Arruda, Rosane Mansan Almeida, Yanna Karla de Medeiros Nóbrega, Maiara dos Santos Carneiro, Tanise Vendruscolo Dalmolin

**Affiliations:** 1Laboratório de Microbiologia e Imunologia Clínica (LabMIC), Departamento de Farmácia, Faculdade de Ciências da Saúde, Universidade de Brasília (UnB), Brasília 70910-900, DF, Brazil; rebecarubens9@gmail.com (R.S.R.); isabelarruda21@gmail.com (I.d.S.A.A.); rmansan@gmail.com (R.M.A.); yannanobrega@gmail.com (Y.K.d.M.N.); 2Centro de Medicina Laboratorial, Farroupilha 95170-476, RS, Brazil; maiaracaneiro7@gmail.com

**Keywords:** antimicrobial susceptibility, polymyxins, Gram-negative bacteria, polymyxin resistance

## Abstract

Antimicrobial resistance is known to be one of the greatest global threats to human health, and is one of the main causes of death worldwide. In this scenario, polymyxins are last-resort antibiotics to treat infections caused by multidrug-resistant bacteria. Currently, the reference test to evaluate the susceptibility of isolates to polymyxins is the broth microdilution method; however, this technique has numerous complications and challenges for use in laboratory routines. Several phenotypic methods have been reported as being promising for implementation in routine diagnostics, including the BMD commercial test, rapid polymyxin NP test, polymyxin elution test, culture medium with polymyxins, and the Polymyxin Drop Test, which require materials for use in routines and must be easy to perform. Furthermore, Sensititre^®^, molecular tests, MALDI-TOF MS, and Raman spectroscopy present reliable results, but the equipment is not found in most microbiology laboratories. In this context, this review discusses the main laboratory methodologies that allow the detection of resistance to polymyxins, elucidating the challenges and perspectives.

## 1. Introduction

The antimicrobial resistance has emerged as a major threat to human health in the 21st century, and is one of the leading causes of death worldwide [1,2]. Along with the increase in antibiotic resistance, polymyxins (colistin and polymyxin B) have been reintroduced for clinical use as valuable therapeutic options, through new formulations and dosage regimens that have considerably reduced the toxicity previously attributed to this class of antimicrobials [3,4].

In recent years, new antimicrobials have been approved for clinical use against multidrug-resistant Gram-negative bacteria, but most do not have activity against all resistance mechanisms and are not available in many parts of the world. Therefore, polymyxins are considered one of the last therapeutic alternatives and are often used as first-line therapy for treating infections caused by multidrug-resistant microorganisms, particularly the Gram-negative bacilli resistant to carbapenems [5].

It was believed that the mechanisms of resistance to polymyxins were mediated by chromosomal mutations, which led to the modification of lipid A (a component of bacterial lipopolysaccharide) through cationic substitutions, to reduce the polymyxin outer membrane interaction [6]. However, in 2015, Liu and collaborators described for the first time the polymyxin resistance mediated by a gene called mobile colistin resistance (*mcr-1*) with plasmid localization. This discovery changed the scenario of polymyxin resistance, due to the possibility of horizontal transfer and the high dissemination of this gene, becoming a major global concern for public health, and consequently, limiting the therapeutic options available [7,8,9].

The *mcr* gene is present worldwide in 72 countries with a high prevalence in animal specimens, and ten *mcr* genes have been described in the literature in eleven species of *Enterobacterales*, with the most prevalent being *Escherichia coli*, *Klebsiella pneumoniae*, and *Salmonella* spp. [7]. In addition, the *mcr-1* gene is the variant with the highest number of descriptions in the scientific literature and 26 subtypes have already been identified (*mcr-1.2* to *mcr-1.27*), which undergo point mutations that lead to small nucleotide changes, giving rise to gene variants [10]. The *mcr* gene encodes a protein that is homologous to the phosphoethanolamine (PEtN) transferase enzyme involved in the lipopolysaccharide modification pathway [7].

It is well-known that there has been a worldwide increase in resistance to polymyxin due to its increased use in clinical practice. Given the current situation, the detection of polymyxin-resistant isolates is becoming increasingly crucial for the correct treatment [11].

Assessing the in vitro susceptibility of polymyxins is fraught with complications, mainly due to their cationic properties, the low diffusion of the polymyxins in agar, the occurrence of heteroresistance in many species, and the adsorption of polymyxins on microtiter plates [11,12]. Currently, the reference test for assessing the susceptibility of isolates to polymyxins is the broth microdilution (BMD) method, which is highly reproducible, reliable, and can be automated. However, the technique is laborious and the manual preparation of antibiotic solutions can lead to significant errors [13].

In this context, this review discusses the main laboratory methods for detecting polymyxin resistance (Figure 1).

## 2. Phenotypic Methods

### 2.1. Broth Microdilution Method and Commercial Tests

BMD is the recommended method for determining the Minimum Inhibitory Concentration (MIC) of polymyxins, according to ISO 20776-1 [14] established by the European Committee on Antimicrobial Susceptibility Testing (EUCAST). The method consists of using different concentrations of polymyxins tested against clinical isolates with concentrations previously determined on microtiter plates [13].

According to the EUCAST, BMD should be carried out using cation-adjusted Mueller–Hinton Broth (CA-MHB) culture medium due to its precise and standardized composition, which ensures consistent results and makes it possible to compare different studies and laboratories. The presence of Ca^2+^ ions in the culture medium is crucial to facilitate the interaction between the antibiotic and the sample, maintaining the ideal concentration. In addition, pure polyester microtiter plates are used without the addition of additives, and it is preferable to use polymyxin powders with sulphated salts. Due to the positively charged chemical structure of polymyxins and their ability to adhere to polystyrene microtiter plates (they can be negatively charged on their surface), the use of glassware to neutralize the adhesion effect is recommended [13,15,16].

However, the use of BMD can bring challenges to the laboratory routine, as well as potential mistakes when performed manually, without the aid of automated systems [17]. BMD is considered a time-consuming test and requires meticulous attention, as well as materials that are difficult to find in routine microbiology laboratories [11].

To mitigate this problem, some companies have developed products designed to detect susceptibility to polymyxins in a more simplified way, without the need for complex equipment. One example of these commercial tests is Policimbac^®^ (Probac do Brasil, São Paulo, Brazil), which uses a microtiter plate containing lyophilized polymyxin B to determine the MIC in Gram-negative strains. Policimbac^®^ consists of a plastic panel with twelve wells, where wells 1 to 10 contain dehydrated CA-MHB with decreasing concentrations of lyophilized polymyxin B (64 to 0.125 mg/L) [18].

Policimbac^®^ was evaluated against 110 Gram-negative isolates (87 *Enterobacterales*, 17 *Acinetobacter* spp. and 6 *Pseudomonas aeruginosa*), including isolates of animal and human origin. The results obtained were 100% Categorical Agreement (CA), which represents the same susceptibility category when comparing Policimbac^®^ with BMD (reference test), but only 16.4% Essential Agreement (EA), which represents the agreement of the MIC (+/− 1log) between the two methods tested. When compared to the reference test, the commercial test showed higher MICs, which may have been because there was no resuspension of the lyophilized polymyxin B, causing the wells to have lower concentrations of polymyxin and consequently increasing the MIC [18].

Rocha et al. analyzed 51 isolates of *K. pneumoniae* against Policimbac^®^ and obtained CA rates of 98.04%. However, the EA was only 31.37%, which was attributed to the fact that the strains analyzed had counts 1 to 6 logs higher than those obtained by the BMD method. This suggests that Policimbac^®^ was not as accurate in correctly identifying these strains when compared to the reference method. Therefore, despite the high overall sensitivity, the low performance in analytical specificity may be an obstacle to the appropriate clinical use of Policimbac^®^ in certain scenarios [19].

Other tests for assessing susceptibility to polymyxins are available on the market, such as ComASP^®^ (Liofilchem, Roseto degli Abruzzi, Italy)—formerly SensiTest™ Colistin—which is a compact plate containing colistin in seven dilutions (0.25–16 mg/L), allowing up to four samples per plate to be tested. The plate is incubated at 36 ± 2 °C and, after the incubation period (16–20 h), it is possible to observe the growth in the wells with the naked eye and to determine the MIC. UMIC^®^ (Biocentric, Collingswood, NJ, USA), on the other hand, is based on the same principle as ComASP^®^ and is a plate compacted with colistin at concentrations of 0.0625 to 64 mg/L. The test comes with a small box that keeps the sample in the ideal incubation atmosphere and the result can be seen with the naked eye. The results obtained by these methods achieved acceptable CA rates of 95.9% for ComASP^®^ and 93.8% for UMIC^®^. However, the EA values were 81.4% for ComASP^®^ and 78.4% for UMIC^®^ (acceptable EA values should be greater than 90%) [20].

The MICRONAUT MIC-Strip Colistin (MERLIN Diagnostika GmbH, Bornheim, Germany) is a broth microdilution system used to determine the colistin MIC for *Enterobacterales*, *Pseudomonas aeruginosa*, and *Acinetobacter baumannii* groups [21]. This method includes a 96-well plate composed of eight removable plastic MIC strips with 12 wells each (colistin concentration range 0.0625 to 64 mg/L). A quantity of 50 µL of bacterial suspension is homogenized in 11.5 mL CAMHB, followed by inoculation of each well with 100 µL of this prepared suspension. After incubation for 18–22 h at 35–37 °C, MICRONAUT-MIC strip evaluation is perform visually using a mirror [22].

A comparative study of different techniques found that the MICRONAUT MIC-Strip Colistin was strongly correlated with the reference MIC, with sensitivity and specificity of 100%, with CA and EA >90%, and no Very Major Error (VME) or Major Error (ME). This demonstrates that the technique can be used for reliable detection [22].

When two commercially available BMD colistin tests were compared—ComASP^®^ and MICRONAUT MIC-Strip Colistin—for nonfermenting rods, the tests indicate *P. aeruginosa* CA of 98.0% and *A. baumannii* CA of 85.7% [21]. MICRONAUT MIC-Strip Colistin had 98.5% sensitivity, 99.5% specificity, 1.5% VME, and 0.5% ME against *A. baumannii* and *Enterobacterales* isolates. This test is recommended as an alternative to BMD for colistin susceptibility testing [23].

VITEK 2^®^ COMPACT (BioMérieux, St. Louis, MO, USA) and Phoenix™ M50 (Becton Dickson Diagnostics, Franklin Lakes, NJ, USA) are the most widely used microbial identification and drug susceptibility analysis systems in the world [24,25]. The performance of these systems was tested against polymyxins in 132 strains of *E. coli* and 83 strains of *K. pneumoniae*, including strains positive for the *mcr-1* and *mcr-8* genes. The systems exhibited excellent CAs and EAs and no false-resistant results for *E. coli* isolates. However, the systems exhibited unacceptable rates of CAs, EAs, and false-susceptible results, especially with *K. pneumoniae* isolates. Furthermore, the systems did not demonstrate good performance for detecting strains with the *mcr-8* gene [24].

Zhu et al. reported that the VITEK 2 system yielded a high VME (25.5%) in 55 *mcr-1*-positive *E. coli* isolates, while Phoenix had an excellent CA (100%) and no ME or VME. Phoenix presented satisfactory results. However, VITEK presented unacceptable errors. The automation achieved the expected results, being reliable for *K. pneumoniae*, while for *Enterobacter* spp., *Acinetobacter* spp., and *Pseudomonas* spp. the equipment did not meet expectations, with many errors in relation to the reference technique. Despite being systems widely used in laboratory routines, they still present results that often must be verified using BMD, making the technique unreliable for testing susceptibility to polymyxins [25].

A fully automated test is Sensititre^®^ (ThermoFisher Diagnostics, Waltham, MA, USA), which contains a plate with wells containing different concentrations of antibiotics. A specific amount of inoculum is added to each well and the plate is incubated at 34–36 °C with humidity control for 18–24 h. Research has shown that the method showed a CA of 97.8% when compared to the reference method. In addition, Sensititre^®^ exhibited a significantly lower error rate compared to UMIC^®^, which showed a VME rate of 11.3%, while Sensitititre^®^ recorded only 3%, thus demonstrating greater safety in the use of this test compared to the others [26].

### 2.2. Rapid Polymyxin NP Test

The Rapid Polymyxin NP Test was developed by Nordmann and Poirel with the aim of obtaining faster (≤4 h) and more accurate results, making it possible to implement it in clinical laboratories, especially in low-resource settings, where the access to antibiotic powders for BMD is limited, facilitating treatment decisions and infection control. The Rapid Polymyxin NP Test is a colorimetric method based on the metabolization of carbohydrates, with a consequent formation of acids that changes the color of the pH indicator [11,27,28].

The test is considered positive, i.e., resistant to polymyxins, if there is bacterial growth in the presence of polymyxin. The test well then changes its color to yellow, indicating carbohydrate/glucose metabolism. The test is considered negative, that is, susceptible to polymyxins, if there is no bacterial growth in the presence of polymyxin. In this case, the color does not change and remains orange [11].

The Rapid Polymyxin NP Test was tested for polymyxin B and colistin, showing positive results for both, with good sensitivity (ranging from 92 to 100%) and specificity (greater than or equal to 90%). Therefore, the NP test is ideal as a screening methodology, as well as being able to be used in countries that face endemic spread of resistance (carbapenemase producers) [11,27,28,29,30,31,32,33].

The Rapid Polymyxin NP Test for *Enterobacter* spp. isolates showed limitations, with sensitivity and specificity of 25 and 100%, respectively [28]. The presence of heteroresistant subpopulations may be the cause of the altered sensitivity of the test [28,31].

The verification of the Rapid Polymyxin NP Test can be optimized using Enzyme Linked Immuno Sorbent Assay (ELISA) at a wavelength of 430 nm. The Receiver Operator Characteristic (ROC) curve, which is used to represent the relationship between sensitivity and specificity of a quantitative diagnostic test, was used to define threshold values for each species: *K. pneumoniae* demonstrated a threshold value of 1.85; for *Enterobacter* the absorption threshold was 1.82; and for *E. coli* the threshold was determined at a value of 1.77. All the samples that showed absorbance values above the threshold were considered resistant and below the susceptibility threshold. Complementing the reading of the results showed a sensitivity of 94% and specificity of 95%, which is reasonable for complementing the NP test, and increased the objectivity of the results [34].

It was also tested against *Pseudomonas* isolates, but with a change in the pH indicator to bromocresol purple, where the change in color from yellow to purple/violet indicates bacterial growth. As well as being quick to obtain isolates, the test showed 100% sensitivity and 95% specificity [35]. When isolates of *A. baumannii* were analyzed, the Rapid Polymyxin NP Test did not show good sensitivity (41.2%) and specificity (86.1%). More objective and sensitive methods, or changes to the indicator, could provide more reliable results [36]. The RapidResa Polymyxin *Acinetobacter* NP^®^ Test (Liofilchem, Roseto degli Abruzzi, Italy) uses resazurin (cell viability indicator) and showed a sensitivity of 96% and specificity of 97% [33].

The Rapid Polymyxin NP Test can also be performed directly from blood cultures, showing good performance, and is easy to implement in laboratories [31,37].

### 2.3. Polymyxin Broth Disk Elution Test

To try to solve the problem of the assessment limits of the polymyxin susceptibility testing and to facilitate the laboratory routine with tests as accurate as BMD, but that use available materials and are easy to perform, the alternative method of Colistin Broth Disc Elution (CBDE) was proposed [38].

CBDE is based on the elution of colistin disks in glass tubes containing CA-MHB, generating final concentrations of 0 (growth control), 1, 2, and 4 mg/L [17]. Subsequently, the standardized bacterial suspension is added, and the result is interpreted according to the established breakpoints [39].

Kanzak et al. evaluated the compatibility, error rates, and its use in the laboratory routine of 89 multidrug-resistant *K. pneumoniae* strains and 5 *E. coli* strains against the CBDE method compared to the reference method. The study obtained 100% CA, demonstrating that the performance of the CBDE test is very good when compared to the reference method [40].

Another multicenter study conducted comparative tests between CBDE and BMD and found similar results for various species, with a CA of 98.6% for *Enterobacterales*, 99.3% for *P. aeruginosa*, and 93.1% for *Acinetobacter* spp. However, specifically for *Acinetobacter* spp., a VME rate of 5.6% and ME rate of 3.3% were observed. ME indicates that the bacterium resistant to the new method is susceptible to the reference method. Although this is a serious error, it may not have as immediate and severe an impact as a VME. These higher numbers of VME and ME may indicate the need to re-evaluate the use of this technique for *Acinetobacter* [39].

Simner et al. conducted a comparative study using 121 retrospective clinical isolates, 45 prospective clinical isolates, and 6 *E. coli* isolates positive for the *mcr-1* gene. The results were like those of the studies, with CA of 98% and EA of 99%. However, the study found errors in the *mcr-1* gene-producing strains where there was a variation in MIC of 2 mg/L for CBDE and 4 mg/L for BMD, changing the categorization of the isolate from susceptible to resistant to colistin. Therefore, it is recommended that when results with an MIC of 2 mg/L are obtained for CBDE, it should be confirmed through BMD in addition to assessing the presence of the *mcr* gene in these isolates [17].

The challenge of detecting polymyxins has led to the need to modify CBDE to make it increasingly effective and useful. In view of this, a study by Dalmolin et al. reduced the volumes used by creating the Colistin Broth Microelution (1 mL) and Colistin Microelution (200 µL) methods on 68 isolates of *Enterobacterales* and 17 non-fermenting Gram-negative bacilli. The results for *Enterobacterales* were satisfactory; however, as reported by Humphries et al., for non-fermenting Gram-negatives, the results were not satisfactory, with high values for ME and VME [39,41].

A promising study conducted by Cielo et al. investigated the elution of polymyxin B, unlike most tests, which focus on the study of colistin. The analysis involved 196 *Enterobacterales*, of which 45.9% showed resistance to polymyxin B. The results were remarkable, with a CA of 99.5% compared to the reference method, and showed 0% for ME and only 1.11% for VME [42].

The detection of strains that produce the *mcr* gene is a major concern, as the broth elution technique does not provide reliable results for bacteria in this specific condition [17]. The structure of the catalytic site of the MCR-1 enzyme is composed of a zinc-dependent metalloprotein and the addition of a chelator such as ethylenediaminetetraacetic acid (EDTA) could reduce the MIC of polymyxins in strains expressing the *mcr* gene [43,44,45]. Thus, a study conducted by Fenwick et al. evaluated the CBDE method with the addition of 0.5 M EDTA for strains producing the *mcr* gene. The results obtained from the CBDE + EDTA method were satisfactory and showed a sensitivity of 100% and specificity of 94.3% for *Enterobacterales* and *P. aeruginosa* strains [46].

Unfortunately, both CBDE and adaptation with the addition of EDTA are not advisable for the detection of *Acinetobacter*, as mentioned in previous studies, and further studies are needed to detect polymyxin resistance in this specific species [39,41,46]. The CBDE methodology is a useful approach with promising potential to be incorporated into laboratory practice, due to its low cost and ease of obtaining materials [42]. However, it is not recommended in the case of isolates expressing the *mcr* gene, for which the EDTA-adapted technique is more suitable [45].

### 2.4. Medium Culture with Polymyxin

Alternative methods for detecting polymyxin resistance based on a culture medium containing polymyxins have been studied as the need for their implementation increased. The focus of many researchers today is to provide a method that can replace or even complement the diagnosis of the reference method, since it is time-consuming and requires many materials [30,47,48].

The SuperPolymyxin™ (ELITech Group, Puteaux, France) medium is based on the eosin methylene blue (EMB) culture medium, adding 3.5 mg/L of polymyxins, along with 10 mg/L of daptomycin (to prevent the growth of *Streptococcus* and *Staphylococcus*) and 5 mg/L of amphotericin B (to prevent fungal growth). The bacterial suspension is standardized to an optical density of 0.5 McFarland (~10^8^ CFU/mL) and plated onto the medium and then incubated at 37 °C, for approximately 24 h. The minimum detection limit for the SuperPolymyxin™ medium was 1 × 10^3^ CFU/mL, and bacterial isolates with growth above or equal to this were resistant to polymyxins. The culture medium showed 100% sensitivity and specificity [49].

A study carried out on stool samples (*n* = 1495) to evaluate the SuperPolymyxin™ medium showed 71.1% sensitivity and 88.6% specificity. The test is suitable for detecting colistin resistance in fecal samples, but a high proportion of susceptible isolates were reported to have grown in the culture medium, requiring confirmation by another technique [48]. Other studies have shown similar results, with values of 82 to 100% for sensitivity and 85 to 97% for specificity [30,50,51].

In addition, there is a challenge in assessing the susceptibility of *Enterobacter* spp., with a reduction in sensitivity (77.3%) due to their tendency to present heterogeneous populations [52].

Colistin Agar Spot is a method that has been increasingly developed as an alternative to the reference method, like the SuperPolymyxin™ medium. The method is based on the dilution of polymyxins in Mueller–Hinton culture medium, following specific concentrations (2.0 mg/L and 3.0 mg/L) to cover the interpretative breakpoints. The bacterial suspensions are standardized at the optical density of 0.5 McFarland and then streaked on the prepared culture medium. The plates are then incubated at 35 °C for 16 to 18 h. To analyze the results, a strain is considered susceptible if no colony growth is observed and resistant if >1 colony growth is observed. Two hundred and seventy-one (271) isolates of Gram-negative bacteria were tested against the Agar Spot method and the researchers obtained a better CA (95.4%) at the colistin concentration of 3 mg/L. The method was satisfactory for *P. aeruginosa*, *Acinetobacter* sp., and *Enterobacterales* [53].

Escalante and collaborators used a modification of the Agar Spot methodology to make a phenotypic identification of the *mcr* gene with EDTA (1 mM). The study presented 96.7% sensitivity and 83.3% specificity, demonstrating the efficiency of the method in differentiating MCR-producing colistin-resistant enterobacteria from those with chromosomal resistance mechanisms [54].

CHROMagar™ COL-APSE (Chromagar, Paris, France) is another alternative culture medium that was developed to be selective in detecting colistin resistance in *Acinetobacter*, *Pseudomonas*, *Stenotrophomonas*, and *Enterobacterales* strains. CHROMagar COL-APSE presented results like those of the SuperPolymixin medium for identifying colistin-resistant microorganisms but has greater sensitivity in the detection of MCR-producing *Enterobacterales*, in addition to providing presumptive chromogenic identification [47]. A study that evaluated some culture media demonstrated that CHROMagar™ COL-APSE had a sensitivity of 82.05% and a specificity of 66.67% and is not recommended for routine use in laboratories [55].

CHROMID^®^ Colistin R Agar (BioMérieux, Marcy-l’Étoile, France) is also a selective chromogenic medium that was developed for the detection of colistin resistance in Gram-negative bacteria, more specifically for *Enterobacterales*. The medium presented a sensitivity of 84.9% and a specificity of 100%, in addition to being able to inhibit the growth of Gram-positive bacteria and fungi [51]. García-Fernández et al. evaluated the performance of this medium for screening colistin-resistant *Enterobacterales* from stool samples and rectal swabs, presenting 100% specificity and sensitivity of 88.1%. Therefore, the CHROMID^®^ Colistin R Agar culture medium proves to be very sensitive and specific for the detection of colistin-resistant *Enterobacterales*, including those that carry the *mcr-1* gene [56].

Sekyere et al. recommend that the Rapid Polymyxin NP Test and/or culture media be used in under-resourced laboratories due to their lower cost as initial screening tools [57].

### 2.5. Polymyxin Drop Test

Initially, the Drop Test was developed to test defensins against *Brucella* isolates. To date, it has been studied and improved so that it can be used in routine laboratories to evaluate the resistance to polymyxins. The test is based on the deposition of a single drop of polymyxin solution (16 mg/L) on a Mueller–Hinton agar plate inoculated with the isolate of interest at 0.5 McFarland [53]. The polymyxin solution, which can be made by diluting the antibiotic powder or by eluting discs containing polymyxins, has a concentration of 16 mg/L, as it has been proven that the solution performs better at this quantity. The plates are left to rest for 15 min at room temperature and subsequently incubated at 35 °C for 16 h to 18 h. After the incubation period, the isolates are considered susceptible when there is the presence of an inhibition zone with well-defined edges, regardless of the diameter, and are considered resistant when there is no presence of a halo around the drop or when there is the presence of colonies in the zone of inhibition [53,58].

The Drop Test was evaluated against 190 isolates of *Enterobacterales* (*K. pneumoniae* and *E. coli*) and 119 *P. aeruginosa* and presented CA of 100%, with no ME or VME detected for *Enterobacterales*. The CA for *P. aeruginosa* strains was 99.2%, where only one VME was observed. The Drop Test is an alternative method for testing the antimicrobial susceptibility of colistin against *K. pneumoniae*, *E. coli*, and *P. aeruginosa*. However, susceptible isolates of *P. aeruginosa* are suggested to be confirmed by BMD. Furthermore, the method has the advantages of not requiring additional equipment and allowing the testing of numerous isolates in a short period of time [59].

Another study evaluated the performance of the Drop Test in detecting the resistance to polymyxin B among *Enterobacterales* and non-fermenting Gram-negative bacilli resistant to carbapenems. In the study, 715 isolates were tested against a drop of polymyxin B at concentrations of 2, 4, and 8 mg/L, with the drop with 4 mg/L being the most accurate. The result was 95.5% CA for *Enterobacterales*, more specifically for *K. pneumoniae*. Therefore, the Drop Test is an easy and quick test to issue a response to detect resistance to polymyxin, thus accelerating the process of therapeutic intervention [60].

## 3. Molecular Methods

The advantages of molecular tests are rapid and accurate detection of resistance mechanisms, as well as the automated analysis of many samples. Furthermore, the molecular tests can detect resistance before its actual phenotypic expression. However, molecular tests require specialized equipment and expertise, making them expensive and less accessible in laboratories with limited resources. In addition, molecular tests can have limitations to predicting clinical results, since the resistance genotype does not always correlate with phenotypic resistance [12].

The demand for different approaches capable of identifying resistance to polymyxins motivated the investigation of methodological alternatives widely used in molecular biology, such as the polymerase chain reaction (PCR) and its variations—quantitative real-time PCR (qRT-PCR) and multiplex PCR [61,62]. For well-resourced laboratories, molecular biology, especially the multiplex PCR assay, can be directly used on cultures to identify colistin-resistant isolates [57].

qPCR is one of the most common techniques when it comes to quantifying nucleic acids. This technique is applicable to both surveillance cultures and biological samples, allowing the detection of genes even when there are reduced amounts of genetic material [61]. As it is a simple method, several researchers chose to use qPCR to identify the *mcr-1* gene [63,64]. Bontron et al. used qPCR with SYBR Green to detect the *mcr-1* gene in cultured bacteria, as well as in spiked human and bovine fecal samples. The *mcr-1* gene was successfully detected, showing a minimum detection limit of 10^2^ CFU/mL cultured bacteria. This test stood out for its high sensitivity and specificity, for not producing false-positive results, and for demonstrating a satisfactory result [63].

Multiplex PCR represents a valuable tool for overcoming the challenge of amplifying multiple nucleic acid targets in a single reaction. In this method, pairs of primers operate under similar conditions to identify distinct individual targets. The effectiveness of this method is directly linked to the design of the primers and the selected temperature, to avoid unwanted reactivity or reduced sensitivity [62]. The application of multiplex PCR may be particularly interesting in laboratories with limited resources, where genetic analysis is necessary to obtain information about resistance mechanisms. This technique allows for an effective and cost-effective approach to understand the bacterial resistance patterns [65].

Rebelo and collaborators developed a multiplex PCR with four sets of primers to amplify the *mcr-1*, *mcr-2*, *mcr-3*, and *mcr-4* genes, in addition to using the primers originally designed for *mcr-5*. This approach was validated by testing 49 animal-derived *E. coli* and *Salmonella* samples. The results demonstrated complete agreement with the whole genome sequencing data and the method was able to identify the *mcr-1*, *mcr-3*, and *mcr-4* genes, both individually and in different combinations, according to their presence in the test isolates [65].

PCR, Sanger sequencing, and qRT-PCR techniques can also be used to investigate mechanisms of chromosomal resistance to polymyxins linked to mutations in the *pmrA*, *pmrB*, *pmrC*, *pmrK*, *phoP*, and *phoQ* genes [66]. To do this, PCR is used to multiply specific segments of DNA (genes of interest) with primers that bind to the sequences flanking the target gene and DNA polymerase to amplify the chosen segment. Sanger sequencing is subsequently used to interpret the results by identifying variations or mutations [67].

Zhang et al. evaluated 504 clinical isolates of carbapenem-resistant *Enterobacterales* in patients without exposure to polymyxins. A total of 19 (3.8%) isolates resistant to polymyxins were detected, and genetic analysis of *K. pneumoniae* strains revealed the presence of insertion sequence elements, a termination codon, and genetic deletion in the *mgrB* gene, as well as a missense mutation in the *pmrB* gene (T157P). Furthermore, two *E. coli* isolates contained the *mcr-1* gene, and a strain of *Enterobacter cloacae* presented mutations of one or more nitrogenous bases in *mgrB*, which is an alert to pre-existing resistance to polymyxin among isolates resistant to carbapenems [66].

Next-generation sequencing (NGS) can be a useful tool in the surveillance of antimicrobial resistance genes by monitoring their emergence and dissemination. The method can analyze a large volume of DNA in a short space of time and consists of fragmenting the target DNA into small pieces and then attaching adapters to the ends of the fragments. The NGS platform reads the nucleotide sequences of each fragment, being performed in parallel, allowing the reading of millions of fragments simultaneously and enabling identification of genetic variations, mutations, insertions, or deletions [68].

Based on the relevance of this monitoring, Li et al. conducted a comprehensive analysis of transregional and interhost dissemination using complete sequences of 455 *mcr*-bearing plasmids (pMCRs) from 44 countries, along with data regarding the host bacteria and the regions where they were isolated. Fifty-two types of Inc replicons were found, including several fusion plasmids containing two or more types of Inc replicons, which were carried by complex host bacteria. The common observation was the occurrence of antibiotic resistance genes in pMCR, with an average of 3.9 antibiotic resistance genes (ARGs) per plasmid. Based on complete plasmid sequences, epidemic events were evidenced that occurred between different countries, over several years, in different sources and hosts. This suggests the possibility of potential spread of pMCRs between humans, food, animals, and the environment [69].

Furthermore, other gene detection tests can be used as surveillance tools, such as the microarray technique, which consists of a large number of DNA probes that can be designed to specifically bind to specific resistance genes. When the sample DNA is applied to the microarray, it is possible to determine which genes are present by hybridizing the probes [70].

Loop-mediated isothermal amplification of DNA (LAMP) is a technique that amplifies DNA with high efficiency, specificity, and rapidity under isothermal conditions. This method employs a DNA polymerase and a set of four specially designed primers that recognize a total of six distinct sequences on the target DNA [71]. LAMP can be used for the detection of the *mcr-1* gene at a constant temperature of 60–67 °C, and dispenses with the need for expensive thermocyclers used in conventional PCR. Amplification products were detected by electrophoresis, colorimetric indicator, and a lateral flow biosensor. The performance of the method for stool samples surprisingly showed a detection limit 10 times higher than that of PCR, demonstrating that the technique is promising, especially for clinical and resource-poor environments. However, a possible disadvantage is that the LAMP technique may be more susceptible to inhibitors present in samples, compared to PCR, affecting the effectiveness of LAMP amplification [72].

The portable platform (Lab-on-a-Chip—LoC system) can be used with the aim of finding a quick and economical solution to detect the *mcr-9* gene, analyzing the bacteria isolated from clinical and screening samples. The results obtained were promising and the average positive detection times were 6.58 ± 0.42 min on a conventional qPCR instrument and 6.83 ± 0.92 min on the LoC platform. This demonstrates that LAMP can become a promising starting point for the development of a near-patient screening test [73].

## 4. Modern Systems: MALDI-TOF MS and Raman Spectroscopy

Matrix-assisted laser desorption ionization-time-of-flight (MALDI-TOF) mass spectrometry (MS) has been incorporated into the routine of clinical laboratories as it is one of the most modern and revolutionary technologies for microbiological diagnosis, since it is practical, fast, accurate, and economical for detecting microorganisms and determining antibacterial susceptibility [74,75,76]. MALDI-TOF MS features two platforms that are commonly used in clinical laboratories: MALDI Biotyper^®^ (Bruker Daltonics, Billerica, MA, USA) and Vitek MS^®^ (BioMérieux, Marcy-l’Étoile, France) which feature a broad library of microbial mass spectra [76,77]. However, even though it is a quick and simple method, spectrum reproducibility may eventually fail, mainly due to closely related and/or genotypically similar species [78].

Several studies have proposed methodologies using MALDI-TOF MS to determine susceptibility to antimicrobials, including polymyxin, as well as to detect resistance mechanisms [75,78,79]. Some of these methodologies that evaluated the efficiency of MALDI-TOF MS technology in detecting resistance to polymyxins are presented below.

MALDI Biotyper-antibiotic susceptibility test rapid assay (MBT-ASTRA) is based on the evaluation of differences in the protein spectra of the bacteria incubated in the presence and absence of the antimicrobial. The tubes are incubated for a certain time and then protein extraction is performed. During the extraction process, an internal control (RNase B), which has a known concentration, is added to all tubes. Afterwards, the protein extracts are analyzed using a Microflex LT/SH bench-top mass spectrometer (Bruker Daltonics, Inc.) to generate protein mass spectra for each tube (MALDI Biotyper 3.1 software). Finally, the Area Under the Curve (AUC) of the spectrum of the bacteria incubated with the antimicrobial is compared with the AUC of the spectrum of the bacteria incubated without the antimicrobial, generating the index called Relative Growth (RG), which reflects the rate of bacterial growth. An RG value close to “1” indicates that the bacteria is resistant to the antimicrobial tested. On the other hand, RG close to “0” indicates susceptible to the antimicrobial [80].

Giordano et al. evaluated the detection of colistin resistance in 139 *K. pneumoniae* isolates from hospitalized patients for MALDI-TOF MS (Bruker Daltonics, Inc.) using FlexAnalysis v3.0 software, MALDI Biotyper v3.0 software, and ClinProtTools v3.0 software. A custom database was created, and classification algorithm models were generated. The strains were correctly identified by the system, showing recognition capacity of the algorithm based on two manually selected mass peaks in 91.8% of the isolates and cross validation in 87.6%. Colistin-resistant strains were correctly classified in 91% and colistin susceptibility was identified in 73% [78].

The main methodologies capable of detecting resistance to polymyxins by MALDI-TOF MS are related to modifications in lipid A, mainly by additions of cationic groups, such as 4-amino-L-arabinose (L-Ara4N) and/or PEtN, to the lipopolysaccharide of the membrane. The addition of PEtN may occur due to the expression of *mcr*-like, a gene of great epidemiological importance today [9]. In 2016, the first study was published that evaluated the main modifications in lipid A, which could be detected by visualizing specific peaks in a spectrum generated by MALDI-TOF MS performed on a 4800 Proteomics Analyzer (Applied Biosystems, Waltham, MA, USA) [81]. Dortet and collaborators developed a technique called MALDIxin, based on MALDI-TOF MS performed on a 4800 Proteomics Analyzer (Applied Biosystems), and MS data were analyzed using Data Explorer version 4.9. The purpose of MALDIxin is discriminating mechanisms of resistance to polymyxin encoded by chromosomes and plasmids, and the approach achieved rapid (15 min) and accurate detection in samples of *E. coli*, and later in *K. pneumoniae* [82,83]. The method was also tested on *P. aeruginosa* isolates using MALDI biotyper Sirius (Bruker Daltonics, Inc.) and, in this species, specifically the signal that corresponds to lipid A can be masked in some resistant strains. Therefore, the addition of polymyxin during the sample preparation phase can improve the detection of resistant *P. aeruginosa* [84].

MALDIxin was also optimized by calculating the value called the Polymyxin Resistance Ratio (PRR), based on the acquired spectra, and using the MALDI Biotyper Sirius system (Bruker Daltonics). PRR values were calculated by summing the intensities of the lipid A peaks attributable to the addition of PEtN (*m*/*z* 1919.2) and L-Ara4N (*m*/*z* 1927.2) and dividing this number by the intensity of the peak corresponding to native lipid A (*m*/*z* 1796.2). Thus, a PRR of 0 indicates susceptibility to colistin, while a positive value indicates resistance, regardless of whether it is chromosomal or plasmid [85].

Fast Lipid Analysis Technique (FLAT) extraction was tested against strains of *Enterobacter* spp. and *K. aerogenes* using the Bruker Microflex LRF MALDI-TOF MS and presented sensitivity and specificity of 100% and 53.4%, respectively. Furthermore, the method is considered to produce faster results (~1 h after the isolate shows growth in culture) [76].

Calderaro and collaborators developed a Classifying Algorithm Model (CAM) using an Autoflex Speed mass spectrometer and data were analyzed using FlexAnalysis software (version 3.1, Bruker Daltonics) and MALDI Biotyper software (version 3.1.66, Bruker Daltonics). The authors tested three different algorithms: Genetic Algorithm (GA), Supervised Neural Network (SNN), and Quick Classifier (QC). Among them, CAMs based on SNN and GA showed the best performances: Recognition Capability (RC) of 100% and Cross-Validation (CV) values of 97.62% and 100%, respectively [86].

The Direct on Target Microdroplets Growth A (DOT-MGA) methodology is based on adding a volume of CA-MHB with antimicrobial and the same volume of a bacterial suspension to a spot on a hydrophobic plate, forming a microdrop. The plate is incubated for a few hours and then the microdrop is removed from the plate using a tissue. The plate with the dried spots is inserted into the mass spectrometer and analyzed using a MALDI Biotyper system and data are analyzed by MALDI Biotyper 3.1. When the bacteria are identified (score ≥ 1.7) in the spot containing the antimicrobial, the bacteria are considered resistant. On the other hand, when identification does not occur (score < 1.7), the bacteria are considered susceptible [87].

Barth et al. proposed a modification of the DOT-MGA methodology, using the conventional steel plate of the mass spectrometer and replacing the disposable hydrophobic plate (single use), to determine the susceptibility to polymyxin B of 122 bacterial isolates cultivated in solid medium and 117 isolates obtained directly from blood cultures positive for carbapenem-resistant Gram-negative bacilli. Bacterial suspension (0.5 McFarland for *Enterobacterales* and 1.5 McFarland for non-fermenting Gram-negative bacilli) and the same volume of a polymyxin B solution were added to obtain a final concentration of 2 mg/L and then spotted onto a reusable steel Bruker MicroFlex LT mass spectrometer (Bruker Daltonics, Inc.) target plate (6 μL). After sample treatment, the isolate was considered resistant when it was identified by the system, even in the presence of 2 mg/L of polymyxin B, and considered susceptible when there was no identification on the equipment. The adapted DOT-MGA presented 95% and 100% CA considering colonies grown on solid media and directly from positive blood culture bottles, respectively. This result was considered very satisfactory, and an excellent alternative for evaluating susceptibility to polymyxin B, while also reducing the evaluation time to just 1 day. Furthermore, the adaptation had much lower input costs than the original technique, as the conventional steel plate can be reused after washing [75].

Another study developed a new MALDI-TOF MS assay in positive ion mode, “CORE—Colistin Resistant”, that allows quantitative or qualitative discrimination between colistin-susceptible or -resistant strains of *K. pneumoniae* within 3 h using the Autof 1000 MS Mass Spectrometer (Autobio Diagnostics, Zhengzhou, China) and Autof Acquirer version 1.0.55 software and the library v2.0.61. This method may be useful for antimicrobial stewardship and for the detection and control of resistant strains in hospital settings [88].

Inamine et al. carried out an adaptation of the MBT-ASTRA technique using the Microfex LT/SH mass spectrometer (Bruker Daltonics, Inc.). They proposed a manual analysis of the spectra, in which three peaks specific to the bacteria and three peaks referring to the internal control would be selected using other software. This adaptation does not require the use of prototype software from the company Bruker, a manufacturer of mass spectrometers, or the R software with the MALDIquant package (https://link.springer.com/article/10.1007/s42770-023-01014-1, accessed on 30 September 2023), making the method more accessible to laboratories that have the equipment. The results of this study indicated that this technique presented excellent sensitivity for evaluating the susceptibility of *Enterobacterales* to polymyxin B [89].

Raman spectroscopy is a form of vibrational spectroscopy, which involves the measurement of scattering spectra, where each peak generated by the spectrum represents different wavelength positions and intensities. In this way, the Raman spectrum is considered capable of generating a unique fingerprint of a given sample, in addition to providing a view of biological macromolecules (lipids, proteins and nucleic acids). To make the method more versatile, isotropic labeling with deuterium (heavy water—deuterium oxide, D_2_O) is added [90].

A protocol for Raman-based antimicrobial susceptibility testing was established. The minimum metabolism inactivation concentration based on the Raman spectra (R-MIC) was developed to quantify strain susceptibility including tigecycline, polymyxin B, and vancomycin, against *E. coli*, *K. pneumoniae*, *P. aeruginosa*, and *Enterococcus faecium*. In the pre-incubation stage, antibiotics are added 1 or 2 h before adding deuterium and the antibiotic is subsequently incubated with deuterium for another 3 h. Resistant strains were demonstrated to absorb more D_2_O, presenting a higher carbon-deuterium ratio value (a visible carbon-deuterium characteristic band appeared on the Raman spectrum—2040–2300 cm^−1^), while susceptible strains presented a carbon-deuterium ratio value lower than the value cutoff, indicating that these strains were possibly metabolically inhibited. The method was able to examine the antimicrobial susceptibility within 5 h with 100% CA and EA. Further clinical investigations are required to validate and popularize this new method [90].

Furthermore, surface-enhanced Raman spectroscopy (SERS) is a technique that employs nanoparticles as substrates, and it is used to characterize and differentiate colistin-resistant and -susceptible *E. coli* strains based on their distinguished SERS spectral features. This study presented 100% specificity, 99.8% sensitivity, and 100% accuracy [91].

Another study developed a rapid method for the detection of colistin resistance in *E. coli, A. baumannii*, and *P. aeruginosa* based on five Raman spectra of each of the samples and analyzed via the hierarchical cluster analysis method to determine whether the bacteria were resistant. The sensitivity and specificity were 90.9% and 91.1%, respectively. This method can be completed in 1.5 h, suggesting its use as a screening method [92].

Lyu et al. (2023) combined SERS spectroscopy (64 SERS spectra for each strain) and a deep learning algorithm convolutional neural network. This method was demonstrated to be noninvasive, low-cost, operational, and fast-paced, and presented high specificity and sensitivity [93].

The advantages, disadvantages, and equipment needed of the main polymyxin susceptibility tests are summarized in Table 1.

## 5. Conclusions

The resurgence of clinical use of polymyxins has assumed an important role as therapy for infections caused by Gram-negative bacteria that might otherwise be intractable. Given this, it is essential that polymyxins are used in an optimized way to preserve their activity for as long as possible, since antimicrobial resistance is a growing public health problem worldwide, having a significant impact on human health and the economy worldwide.

The challenges of polymyxin susceptibility testing are undeniable and numerous. The reference test has challenges and controversies. The development and improvement of standardized, fast, easy, and low-cost polymyxin susceptibility tests are extremely important for incorporation in clinical microbiology laboratories, especially where there is a shortage of materials.

Several methods to determine the susceptibility to polymyxins are reported as promising for implementation, according to the structure of each clinical laboratory.

There are two different yet supplementary perspectives for the future of polymyxin susceptibility tests: (1) the phenotypical methods, such as the Rapid Polymyxin NP Test and Drop Test, for screening and/or for laboratories with few technological resources, which require methods that can use available materials in the routine and are easy to perform and low-cost; and (2) modern systems, such MALDI-TOF, Sensititre^®^, and molecular tests, to confirm results and understand the polymyxin resistance, and/or for laboratories with technological resources. Despite obtaining reliable results, most routine laboratories lack the required equipment, and such tests are widely used in research. It is expected that, in the future, laboratories will have access to these technologies via the reduction in equipment required and the input costs.

## Figures and Tables

**Figure 1 microorganisms-12-00101-f001:**
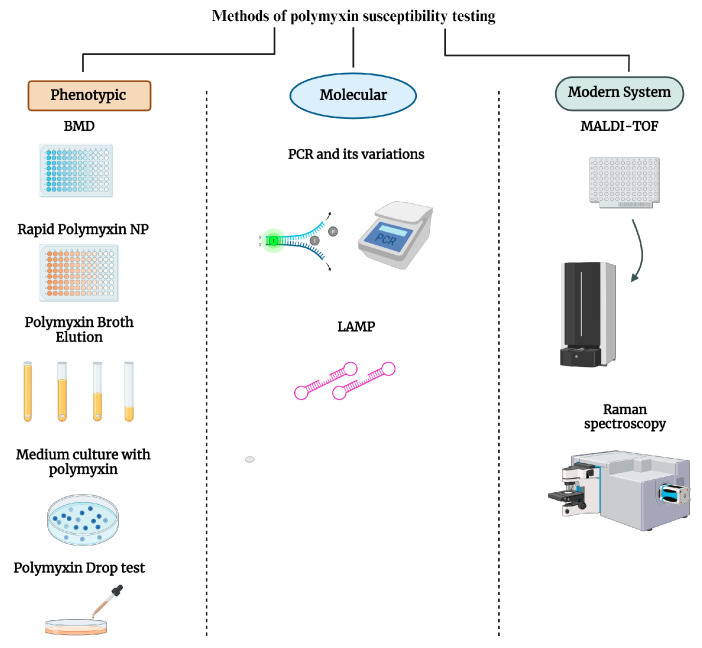
Main methods for detecting polymyxin susceptibility according to the principle. Created with BioRender.com, accessed on 21 November 2023. BMD: broth microdilution; PCR: polymerase chain reaction; LAMP: loop-mediated isothermal amplification of DNA; MALDI-TOF: matrix-assisted laser desorption ionization-time-of-flight (MALDI-TOF) mass spectrometry.

**Table 1 microorganisms-12-00101-t001:** Advantages and disadvantages of main polymyxin susceptibility testing.

Method		Advantages	Disadvantages	Equipment Needed
Phenotypic Method Broth microdilution method	Conventional BMD	Recommended by ISO and EUCAST for determining MIC of polymyxins; Allows comparison between different studies and laboratories (highly reproducible); Reliable and can be automated.	Laborious; Time-consuming and requires meticulous attention; Requires many materials that are difficult to find in routine microbiology laboratories (antibiotic powder).	No
	Policimbac^®^ (Probac do Brasil)	Plate containing lyophilized polymyxin B (it is not necessary to use antibiotic powder).	Higher MICs due to the lack of resuspension of lyophilized polymyxin B.	No
	ComASP^®^ (Liofilchem)	Compact plate (it is not necessary to use antibiotic powder); Allows testing of multiple isolates.	Unacceptable values for Essential Agreement (EA); Lack of resuspension of lyophilized polymyxin.	No
	UMIC^®^ (Biocentric)	Compact plate (it is not necessary to use antibiotic powder); Allows testing of multiple isolates; Small box that keeps the sample in the ideal incubation atmosphere.	Unacceptable values for Essential Agreement (EA); Lack of resuspension of lyophilized polymyxin.	No
	MICRONAUT MIC-Strip Colistin (Merlin)	Compact plate (it is not necessary to use antibiotic powder); Allows testing of multiple isolates.	Lack of resuspension of lyophilized polymyxin.	No
	Automated systems—VITEK^®^ COMPACT (BioMérieux) and Phoenix™ (Becton)	Allows testing of multiple samples; Automated; Fast and easy.	Unacceptable rates of CAs, EAs, and false-susceptible results; Unreliable for polymyxin susceptibility testing; Cost equipment.	Yes
	Sensititre^®^ (ThermoFisher Diagnostics)	Fully automated test; Humidity and temperature control; High concordance with the reference method; Exhibits significantly lower error rates compared to other tests.	Cost of reagents and equipment.	Yes
Phenotypic Method Rapid Polymyxin NP Test		Fast (≤4 h); Performed directly from blood cultures; Good performance; Easy to implement in laboratories; Low cost.	Limitations for *Enterobacter* spp. (heteroresistant subpopulations); *Acinetobacter baumannii* did not show good sensitivity and specificity.	No. Can be optimized using Enzyme Linked Immuno Sorbent Assay
Phenotypic Method Polymyxin Broth Disk Elution Test		Accurate and reliable results; Available materials use in routine; Easy; Low Cost; Good performance.	Limitation in *Acinetobacter* spp. isolates and in strains that express *mcr*-1 gene (addition of EDTA may be necessary).	No.
Phenotypic Method Medium Culture with Polymyxin (SuperPolymyxin medium, Agar Spot, CHROMagar™ COL-APSE, CHROMID^®^ Colistin R Agar)		Can be performed from stool samples and rectal swabs; Efficiency in differentiating MCR-producing colistin-resistant enterobacteria from those with chromosomal resistance mechanisms (Agar Spot + EDTA); Low cost; Easy.	Challenge in assessing the susceptibility of *Enterobacter* spp. (heterogeneous populations); CHROMagar™ COL-APSE presented low sensitivity and specificity.	No
Phenotypic Method Polymyxin Drop Test		Easy and fast; Allows testing of multiple isolates; Low cost; Available materials use in routine.	Challenge in assessing the susceptibility of *P. aeruginosa.*	No
Molecular Method (PCR, qRT-PCR, Multiplex PCR, Sanger Sequencing, Next-Generation Sequencing, Microarray)		Reduced amounts of genetic material (detection limit of 10^2^ CFU/mL); Multiple nucleic acid targets in a single reaction (Multiplex PCR); Identification of genetic variations, mutations, insertions, or deletions (sequencing).	Specific optimization for different genes; Cost of reagents and equipment; Expertise in primer design and bioinformatic.	Yes
Molecular Method (Loop-Mediated Isothermal Amplification of DNA)		High efficiency and specificity; Fast; Dispense the need for expensive thermocyclers used in PCR; Detection limit 10 times higher than PCR.	Susceptible to inhibitors present in clinical samples; Expertise in primer design and bioinformatic.	Yes
Modern Systems (MALDI-TOF MS)		Simple; Fast; Accurate and economical.	Spectrum reproducibility may eventually fail, mainly due to closely related and/or geno-typically similar species. Cost of equipment.	Yes
Modern Systems (Raman spectrometry)		Low-cost; Operational; Fast.	Cost of equipment; Materials that are difficult to find in routine microbiology.	Yes

## Data Availability

Data are contained within the article.

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
