# Peer review of "Challenges in the Detection of Polymyxin Resistance: From Today to the Future"

_microorganisms, 2024, doi:10.3390/microorganisms12010101_

Round 1
Reviewer 1 Report
Comments and Suggestions for Authors
The study is a review so:
- the Authors missed some of the tests BMD e.g.: SensiTest Colistin (Liofilchem), MIC-Strip (Merlin), MIC COL (Diagnostics)
- LAMP method in molecular diagnostics should be included in the text
- Some of the references are missing e.g. Sekyere „Mcr colistin …”, SÄ™kowska „Energence of colistin…”
- what type of MALDI TOF was used Microflex, SMART or other?
- what library version was used?
- Bruker is the name of the company, not analyser
- in the text there are typos.
Author Response
Dear Reviewer,
Thanks for sending the comments regarding our manuscript (microorganisms-2718580). We have considered all comments, and our responses are below. We have also altered the manuscript according to the comments and the revised version contains the modifications highlighted in yellow.
We hope that the manuscript is now suitable to be accepted for publication in Microorganisms.
Sincerely,
Tanise Vendruscolo Dalmolin
Reviewer 1
- The Authors missed some of the tests BMD e.g.: SensiTest Colistin (Liofilchem), MIC-Strip (Merlin), MIC COL (Diagnostics).
The MICRONAUT MIC-Strip (Merlin) test was added in the text.
MIC-COL (Diagnostics) test was not found from articles, then it was not added.
The SensiTest Colistin (Liofilchem) was already in the article with update name - ComASP® (Liofilchem). This information was added in the text.
- LAMP method in molecular diagnostics should be included in the text
LAMP method was added in the text.
- Some of the references are missing e.g. Sekyere „Mcr colistin …”, SÄ™kowska „Energence of colistin…”
The reference Sekyere (mcr colistin resistance gene: a systematic review of current diagnostics and detection methods) was added in the text.
The reference Sekowska (Emergence of colistin-resistant Klebsiella pneumoniae in Poland) was not cited for reporting local data.
- what type of MALDI TOF was used Microflex, SMART or other? - what library version was used?
The type of MALDI-TOF and library version used in the references was added when present in the article.
- Bruker is the name of the company, not analyser
The sentence was modified to MALDI Biotyper® (Bruker, Inc).
- In the text there are typos.
The typos have been corrected.
Reviewer 2 Report
Comments and Suggestions for Authors
I think that the manuscript entitled “Challenges in the detection of the polymyxin resistance: from today to the future” is in principle suited for a publication in Microorganisms, Special Issue “Resistant Bacteria: What Course to Follow?”. It provides an overview of various laboratory methodologies for detecting polymyxin resistance, ranging from traditional techniques to advanced methods like MALDI-TOF MS and Sensititre®, which is particularly pertinent for researchers and practitioners in clinical and microbiological fields. However, I have some comments and concerns.
Major comments:
The manuscript lacks any figures and tables, which may reduce reader interest. Please ensure that the main results and conclusions of the review are presented graphically (in the form of diagrams or illustrations). For individual sections of the manuscript, tables with references and key findings from the reviewed works may be appropriate.
It seems that the authors could have more fully considered modern systems for detecting resistance to polymyxins. In particular, the use of Heavy water (deuterium oxide, D2O)-labeled Raman spectroscopy (https://www.tandfonline.com/doi/full/10.2147/IDR.S404732). Additionally, various commercial systems could be compared, as was done for Vitek 2 and Phoenix M50 (https://www.tandfonline.com/doi/full/10.2147/IDR.S400772).
Overall, I would like to see a final table summarizing the advantages and disadvantages of each method reviewed by the authors of the manuscript. Without this, the presented attempt at a review looks merely like a recounting of known studies.
Minor comments:
Line 64. The abbreviation "LPS" should be expanded upon its first usage. If it is not used more than once, the term should be written in its full form.
Lines 166-197. The text might benefit from a clearer summarization or conclusion that ties together the various findings and their implications for the use of the Rapid Polymyxin NP Test in different scenarios.
Lines 204-205. “(Simner et al., 2018)”. It seems that the authors used a different citation format here.
Line 223. Phrases like "what is striking about the study"could be more formally expressed.
Lines 305-308. “CHROMID® Colistin R Agar is also a selective chromogenic medium that was developed for the detection of colistin resistance in Gram-negative bacteria, more specifically for Enterobacterales, but does not include species of Acinetobacter, Pseudomonas and Stenotrophomonas [41,45]”. I think that CHROMID® is not mentioned in the cited work â„– 41. Please check.
Line 319. "Mueller-Hilton agar" should be "Mueller-Hinton agar."
Lines 347, 373. The abbreviation "qPCR-RT" is more commonly known as "qRT-PCR" (quantitative Real-Time PCR).
Line 354: The phrase "102 cultured bacteria" should specify the unit of measurement (like "102 CFU/mL" or another appropriate unit).
Line 383. The term "mutation missense" should be "missense mutation".
Line 416. The brand names "Bruker Maldi Biotyper®" and "Vitek MS®" should be consistently formatted and capitalized. For example, "Maldi" should be "MALDI."
Lines 434-435: "CR" should be consistently referred to as "RG" (relative growth), as initially introduced.
Line 486. The term "MacFarland" should be corrected to "McFarland."
Author Response
Dear Reviewer,
Thanks for sending the comments regarding our manuscript (microorganisms-2718580). We have considered all comments, and our responses are below. We have also altered the manuscript according to the comments and the revised version contains the modifications highlighted in yellow.
We hope that the manuscript is now suitable to be accepted for publication in Microorganisms.
Sincerely,
Tanise Vendruscolo Dalmolin
Reviewer 2
- The manuscript lacks any figures and tables, which may reduce reader interest. Please ensure that the main results and conclusions of the review are presented graphically (in the form of diagrams or illustrations). For individual sections of the manuscript, tables with references and key findings from the reviewed works may be appropriate.
We agree with this comment.
We added in the text a table (page 15 - line 623) and figure (page 4 - line 81) to increase reader interest.
- It seems that the authors could have more fully considered modern systems for detecting resistance to polymyxins. In particular, the use of Heavy water (deuterium oxide, D2O)-labeled Raman spectroscopy (https://www.tandfonline.com/doi/full/10.2147/IDR.S404732).
Raman spectroscopy was added in the section “Modern systems” together with MALDI-TOF (pages 13-14, lines 587-621).
- Additionally, various commercial systems could be compared, as was done for Vitek 2 and Phoenix M50 (https://www.tandfonline.com/doi/full/10.2147/IDR.S400772).
The reference was added in the text (page 5, lines 160-177). Others comercial systems were compared and added in the text.
-Overall, I would like to see a final table summarizing the advantages and disadvantages of each method reviewed by the authors of the manuscript. Without this, the presented attempt at a review looks merely like a recounting of known studies.
We agree with this comment.
We added in the text a table summarizing main method (page 15 - line 623).
Line 64. The abbreviation "LPS" should be expanded upon its first usage. If it is not used more than once, the term should be written in its full form.
The term was written in its full form (page 2, line 64).
Lines 166-197. The text might benefit from a clearer summarization or conclusion that ties together the various findings and their implications for the use of the Rapid Polymyxin NP Test in different scenarios.
The summarization of the findings was added in the text (page 6, lines 200-204).
Lines 204-205. “(Simner et al., 2018)”. It seems that the authors used a different citation format here.
The citation format was rewritten correctly (page 7, line 235).
Line 223. Phrases like "what is striking about the study" could be more formally expressed.
The sentence was rewritten more formally expressed (page 7, lines 253-256).
Lines 305-308. “CHROMID® Colistin R Agar is also a selective chromogenic medium that was developed for the detection of colistin resistance in Gram-negative bacteria, more specifically for Enterobacterales, but does not include species of Acinetobacter, Pseudomonas and Stenotrophomonas [41,45]”. I think that CHROMID® is not mentioned in the cited work â„– 41. Please check.
Thank you for pointing this out. The references were corrected (page 9, line 343)
Line 319. "Mueller-Hilton agar" should be "Mueller-Hinton agar."
Thank you for pointing this out. The sentence was rewritten (page 9, line 350).
Lines 347, 373. The abbreviation "qPCR-RT" is more commonly known as "qRT-PCR" (quantitative Real-Time PCR).
The sentence was rewritten (page 10, lines 387 and 415).
Line 354: The phrase "102 cultured bacteria" should specify the unit of measurement (like "102 CFU/mL" or another appropriate unit).
The sentence was rewritten (page 10, line 396).
Line 383. The term "mutation missense" should be "missense mutation".
The sentence was rewritten (page 10, line 425).
Line 416. The brand names "Bruker Maldi Biotyper®" and "Vitek MS®" should be consistently formatted and capitalized. For example, "Maldi" should be "MALDI."
The names were formatted throughout the text.
Lines 434-435: "CR" should be consistently referred to as "RG" (relative growth), as initially introduced.
We changed “CR” by “RG” when referring to relative growth (page 12, lines 498-499).
Line 486. The term "MacFarland" should be corrected to "McFarland."
The sentence was rewritten (page 13, line 559).
Reviewer 3 Report
Comments and Suggestions for Authors
This paper reviewed the laboratory methodologies for detecting the polymyxin resistance, mainly on phenotypical resistance techniques.
1. The authors discussed the polymyxin resistance test methods in different title, such as broth microdilution method tests, rapid polymyxin NP test, as well as molecular biology, MALDI-TOF MS techniques. It would be better summarizing these methods according the principle, for example, phenotypical resistance test, genotypical tests, etc.
2. Many phenotypical resistance techniques were discussed. Using a figure or a table to give a brief comparison of these techniques would be better.
3. The molecular biology, antimicrobial resistance gene analysis, were also discussed. The advantages and disadvantages comparing to phenotypical tests should be addressed.
4. What is the perspectives of polymyxin resistance test?
Author Response
Dear Reviewer,
Thanks for sending the comments regarding our manuscript (microorganisms-2718580). We have considered all comments, and our responses are below. We have also altered the manuscript according to the comments and the revised version contains the modifications highlighted in yellow.
We hope that the manuscript is now suitable to be accepted for publication in Microorganisms.
Sincerely,
Tanise Vendruscolo Dalmolin
Reviewer 3
The authors discussed the polymyxin resistance test methods in different title, such as broth microdilution method tests, rapid polymyxin NP test, as well as molecular biology, MALDI-TOF MS techniques. It would be better summarizing these methods according the principle, for example, phenotypical resistance test, genotypical tests, etc.
We added in the text a figure summarizing the methods according the principle (page 3, line 79).
Many phenotypical resistance techniques were discussed. Using a figure or a table to give a brief comparison of these techniques would be better.
We added in the text a table (page 14, line 619) and figure (page 3, line 79) to increase reader interest.
The molecular biology, antimicrobial resistance gene analysis, were also discussed. The advantages and disadvantages comparing to phenotypical tests should be addressed.
The advantages and disadvantages between phenotypic and genotypic tests have been added in the section “Molecular Tests” (page 9, lines 373-379).
What is the perspectives of polymyxin resistance test?
The conclusion was rewritten about perspectives of polymyxin resistance tests (page 18, lines 637-646).
Reviewer 4 Report
Comments and Suggestions for Authors
1. The sentence of Line 20-21 and line 35-36 is very same.
2. Line 43, “mechanisms“ should be replaced by other words.
3. In part of introduction, the logic is not clear.
4. Line 204, literature citation format is not correct.
5. Reference marks are missing in many places.
6. Line 327, punctuation misuse.
7. Each method draws a diagrammatic sketch, which is more intuitive and improves the reference rate of the article.
8. The language needs to be polished, more concise, and the meaning expression is more clear.
9. There is no vision for the future.
10. I did not see the author's opinion.
Comments on the Quality of English Language
1. The sentence of Line 20-21 and line 35-36 is very same.
2. Line 43, “mechanisms“ should be replaced by other words.
3. In part of introduction, the logic is not clear.
4. Line 204, literature citation format is not correct.
5. Reference marks are missing in many places.
6. Line 327, punctuation misuse.
7. Each method draws a diagrammatic sketch, which is more intuitive and improves the reference rate of the article.
8. The language needs to be polished, more concise, and the meaning expression is more clear.
9. There is no vision for the future.
10. I did not see the author's opinion.
Author Response
Dear Reviewer,
Thanks for sending the comments regarding our manuscript (microorganisms-2718580). We have considered all comments, and our responses are below. We have also altered the manuscript according to the comments and the revised version contains the modifications highlighted in yellow.
We hope that the manuscript is now suitable to be accepted for publication in Microorganisms.
Sincerely,
Tanise Vendruscolo Dalmolin
Reviewer 4
The sentence of Line 20-21 and line 35-36 is very same.
Thank you for pointing this out. The sentence from abstract was rewritten (page 1, lines 20-21).
Line 43,“mechanisms“ should be replaced by other words.
The word “mechanisms” refers to “resistance mechanisms” as mentioned in the reference used (https://doi.org/10.1007/s10096-020-04053-w) and others articles.
In part of introduction, the logic is not clear.
The introduction is organized in the following sequence:
- antimicrobial resistance
- polymyxins as one of the last therapeutic
- mechanisms of resistance to polymyxins (chromosomal and plasmidial – mcr genes)
- detection of polymyxin-resistant
Polymyxin Susceptibility testing and complications.
Line 204, literature citation format is not correct.
The citation format was rewritten correctly (page 6, line 231).
Reference marks are missing in many places.
The references were added.
Line 327, punctuation misuse.
The punctuation was modified.
Each method draws a diagrammatic sketch, which is more intuitive and improves the reference rate of the article.
We added a figure (page 3, line 79) summarizing the methods to increase reader interest.
The language needs to be polished, more concise, and the meaning expression is more clear.
The language was corrected in certain places in the text.
There is no vision for the future. I did not see the author's opinion.
The conclusion was rewritten about perspectives of polymyxin resistance tests (page 18, lines 637-646).
Reviewer 5 Report
Comments and Suggestions for Authors
The paper offers excellent up to date overview of contemporary methods for detection of the polymyxin resistance. It is well structured easy to follow.
I lack tables and figures or diagrams that would make it much easier to quickly get an overview. Please add a figure describing most important aspects of each method, including method type, cost, availability, level of staff training needed, if it is recommended in guidelines etc. A table should offer all these information on a quick look.
Line 66 - reference should be added
please check in detail that all claims are appropriately referenced.
Author Response
Dear Reviewer,
Thanks for sending the comments regarding our manuscript (microorganisms-2718580). We have considered all comments, and our responses are below. We have also altered the manuscript according to the comments and the revised version contains the modifications highlighted in yellow.
We hope that the manuscript is now suitable to be accepted for publication in Microorganisms.
Sincerely,
Tanise Vendruscolo Dalmolin
Reviewer 5
I lack tables and figures or diagrams that would make it much easier to quickly get an overview. Please add a figure describing most important aspects of each method, including method type, cost, availability, level of staff training needed, if it is recommended in guidelines etc. A table should offer all these information on a quick look.
We agree with this comment. We added in the text a table (page 14, line 619) and figure (page 3, line 79) to increase reader interest.
Line 66 - reference should be added. Please check in detail that all claims are appropriately referenced.
The reference was added in the text (page 2, lines 66-69).
Round 2
Reviewer 1 Report
Comments and Suggestions for Authors
I need clarification. The author writes that "The reference Sekowska (Emergence of colistin-resistant Klebsiella pneumoniae in Poland) was not cited for reporting local data." It' s not understandable, because the article of Depka et al. is about local data (from the same medical center) yet later on it was cited.
Author Response
Dear Reviewer,
Thank you very much for taking the time to review this manuscript (microorganisms-2718580). We have altered the manuscript according to the comments and the revised version contains the modifications highlighted in yellow.
We hope that the manuscript is now suitable to be accepted for publication in Microorganisms.
Sincerely,
Tanise Vendruscolo Dalmolin
Reviewer 1:
I need clarification. The author writes that "The reference Sekowska (Emergence of colistin-resistant Klebsiella pneumoniae in Poland) was not cited for reporting local data." It' s not understandable, because the article of Depka et al. is about local data (from the same medical center) yet later on it was cited.
The Sekowska (The Sekowska reference did not present sensitivity and especificity data about the tests. Therefore, it is difficult to compare with other studies and our study, and the reference was not added in the text) reference did not present sensitivity and especificity data about the tests. Therefore, it is difficult to compare with other studies and our study, and the reference was not added in the text.
Reviewer 2 Report
Comments and Suggestions for Authors
The authors have significantly revised the manuscript and made the necessary corrections. In its current form, the manuscript can be recommended for publication.
Author Response
Dear Reviewer,
Thank you very much for taking the time to review this manuscript (microorganisms-2718580).
We hope that the manuscript is now suitable to be accepted for publication in Microorganisms.
Sincerely,
Tanise Dalmolin
Reviewer 3 Report
Comments and Suggestions for Authors
Please formate table 1 to make it more readable.
Author Response
Dear Reviewer,
Thank you very much for taking the time to review this manuscript (microorganisms-2718580). We have altered the manuscript according to the comments and the revised version contains the modifications highlighted in yellow.
We hope that the manuscript is now suitable to be accepted for publication in Microorganisms.
Sincerely,
Tanise Dalmolin
Reviewer 3:
Please formate table 1 to make it more readable.
We agree with this coment. The Table 1 was formated.
Reviewer 4 Report
Comments and Suggestions for Authors
1. Line 156-164, line 165-173, line 452-459, and line 460-466, These paragraphs cite only one document each, a little less. This problem also exists in many places in the article.
2. Line 170,"technique Despite" has less punctuation marks in the middle
3. Line 589,Please verify whether "2" in "D2O" is wrong.
4. Parts 2-8 should belong to different categories, corresponding to the method classification in Figure 1. At the same time, the contents in Table 1 are also classified according to the method of Figure 1, and the contents can be further refined.
5. Ideas about the future should be the most attraction, but they was less. So it should be a good way to imagine the future of each approach.

Author Response
Dear Reviewer,
Thank you very much for taking the time to review this manuscript (microorganisms-2718580). We have altered the manuscript according to the comments and the revised version contains the modifications highlighted in yellow.
We hope that the manuscript is now suitable to be accepted for publication in Microorganisms.
Sincerely,
Tanise Vendruscolo Dalmolin
Reviewer 4:
Line 156-164, line 165-173, line 452-459, and line 460-466, These paragraphs cite only one document each, a little less. This problem also exists in many places in the article.
These highlighted paragraphs discuss data from specific studies. Therefore, these paragraphs was cited only one article. We cited more references in some paragraphs, but others was cited only one.
Line 170,"technique Despite" has less punctuation marks in the middle
Thank you for pointing this out. The punctuation was added.
Line 589,Please verify whether "2" in "D2O" is wrong.
Thank you for pointing this out. Heavy water (deuterium oxide, D2O) was corrected in the text.
Parts 2-8 should belong to different categories, corresponding to the method classification in Figure 1. At the same time, the contents in Table 1 are also classified according to the method of Figure 1, and the contents can be further refined.
We agree with the reviewer. We changed the categories in the text, according to Figure 1. Furthermore, the method classification was added in the Tabla 1.
Ideas about the future should be the most attraction, but they was less. So it should be a good way to imagine the future of each approach.
The future of each approach is very difficult. Therefore an overview of the future of detection of the polymyxin resistance has been added in the conclusion. Furthermore, challenges and improvements in techniques were discussed in each section.